# Seroprevalence of Hantavirus among Manual Cane Cutters and Epidemiological Aspects of HPS in Central Brazil

**DOI:** 10.3390/v15112238

**Published:** 2023-11-10

**Authors:** Renata Malachini Maia, Jorlan Fernandes, Luciana Helena Bassan Vicente de Mattos, Luiz Antonio Bastos Camacho, Karlla Antonieta Amorim Caetano, Megmar Aparecida dos Santos Carneiro, Fernando de Oliveira Santos, Sheila Araujo Teles, Elba Regina Sampaio de Lemos, Renata Carvalho de Oliveira

**Affiliations:** 1Hantaviruses and Rickettsiosis Laboratory, Oswaldo Cruz Institute (FIOCRUZ), Rio de Janeiro 21040-360, Brazil; jorlan@ioc.fiocruz.br (J.F.); lucitahelena@hotmail.com (L.H.B.V.d.M.); elba.lemos@gmail.com (E.R.S.d.L.); 2National School of Public Health, Oswaldo Cruz Institute (FIOCRUZ), Rio de Janeiro 21041-210, Brazil; luizantonio.camacho@fiocruz.br; 3Faculty of Nursing, Federal University of Goiás, Goiânia 74605-080, Brazil; karllacaetano@gmail.com (K.A.A.C.); sheila.fen@gmail.com (S.A.T.); 4Institute of Tropical Pathology and Public Health, Federal University of Goiás, Goiânia 74605-080, Brazil; megmar242@gmail.com; 5Biology and Parasitology of Wild Mammals Reservoirs Laboratory, Oswaldo Cruz Institute (FIOCRUZ), Rio de Janeiro 21041-210, Brazil; fernando.oliveira.snts@gmail.com

**Keywords:** hantavirus, hantavirus pulmonary syndrome, sugarcane cutters, rural population, Brazil

## Abstract

Hantavirus pulmonary syndrome (HPS) is a rodent-borne zoonotic disease that is endemic throughout the Americas. Agricultural activities increase exposure to wild rodents, especially for sugarcane cutters. We carried out a survey of the epidemiological aspects of HPS and investigated the prevalence of hantavirus infection in the sugarcane cutter population from different localities in the Brazilian Midwest region. We conducted a retrospective study of all confirmed HPS cases in the state of Goiás reported to the National HPS surveillance system between 2007 and 2017, along with a seroepidemiological study in a population of sugarcane cutters working in Goiás state in 2016, using the anti-hantavirus (Andes) ELISA IgG. A total of 634 serum samples from cane cutters were tested for hantavirus antibodies, with 44 (6.9%) being IgG-reactive according to ELISA. The destination of garbage was the only statistically significant variable (*p* = 0.03) related to the detection of hantavirus IgG (*p* < 0.05). We described the epidemiological profile of reported hantavirus cases in Goiás—a highly endemic area for HPS, and where the seroepidemiological study was conducted. Our results increase our knowledge about hantavirus infections in Brazil and highlight the vulnerability of sugarcane cutters to a highly lethal disease that, to date, has no specific treatment or vaccination.

## 1. Introduction

Hantaviruses belong to the *Hantaviridae* family, which is currently divided into four subfamilies [1]. To date, all hantaviruses reported to infect humans belong to the *Mammantavirinae* subfamily—specifically, the *Orthohantavirus* genus. These viruses are spherical, enveloped, and measure between 80 and 120 nm in diameter. They have glycoprotein projections on their surface and a tri-segmented RNA genome consisting of a small segment (S), a medium segment (M), and a large segment (L) [2,3].

These zoonotic viruses are transmitted by different species of wild and synanthropic rodents. Transmission usually occurs in rural or peri-urban areas with wild rodent infestations, often associated with poor sanitary conditions. Virus particles are excreted through infected rodents urine, feces, and saliva and can be inhaled through aerosols formed from the drying secretions and excreta. Other possible forms of transmission include food contaminated by animal waste, bites, and, more rarely, contact of contaminated hands with the mouth or nose [4,5]. Human infections have been divided into two clinical forms: hemorrhagic fever with renal syndrome (HFRS) [6,7], and hantavirus pulmonary syndrome (HPS), also known as hantavirus cardiopulmonary syndrome (HCPS), which is restricted to the American continent and has a lethality ranging from 40.0% to 70.0% [8,9]. According to data available on the Health Surveillance Secretariat (SVS) website of the Brazilian Ministry of Health, as of 2022, approximately 2050 cases and 808 deaths have been confirmed since 1993 [10].

Brazil is considered the world’s largest sugarcane producer, and the state of Goiás ranks second in national production [11]. In the Midwest region, where Goiás State is located, sugarcane cutting is still the main method of harvesting, increasing the chances of human contact with infected rodents that may be present in the cane fields in search of food. Since agricultural profiles are determining factors for the transmission of hantavirus, this economic activity deserves special attention in the epidemiology of this zoonosis in Brazil [12]. Moreover, about 70.0% of the territory of Goiás is covered by the Cerrado biome, the habitat of the rodent *Necromys lasiurus*, which is the most abundant hantavirus host in this biome and is associated with the Araraquara virus [13]. Considered a biome with endemic areas for hantavirus infection, as of 2022, 123 cases of hantavirus were registered in this federative unit [10].

In this study, we assessed the epidemiological profile of reported hantavirus cases in Goiás State and conducted a cross-sectional seroepidemiological investigation to estimate the prevalence of hantavirus antibodies among manual sugarcane cutters working in five distinct plants. This study is expected to increase our knowledge about this zoonosis in an endemic area and highlight the vulnerability of sugarcane cutters, emphasizing the need for control and prevention actions.

## 2. Materials and Methods

### 2.1. Retrospective Descriptive Study of HPS Cases in Goiás State

Based on the available data from SINAN (Notifiable Diseases Information System), variables were selected to conduct a descriptive analysis of HPS occurrence in Goiás State between 2007 and 2017 [14]. The analysis included the following variables: (i) environment of infection, (ii) confirmed cases per year, (iii) education level, (iv) age group, (v) notification municipality, (vi) race, (vii) sex, and (viii) residential area (Table 1).

### 2.2. Retrospective, Analytical, and Cross-Sectional Seroepidemiological Investigation in a Rural Population in the State of Goiás

Individuals working as manual sugarcane cutters in alcohol and sugar-producing units in five municipalities of the state of Goiás (Figure 1) were voluntarily enrolled in the study conducted between August and September 2016, as previously reported [15].

All information obtained from the sugarcane cutters included in the study was collected by trained research assistants using a questionnaire during interviews that took place in the cane fields. The available variables were included in the present study based on their hypothesized correlation with the epidemiological profile of hantavirus infection.

The prediction variables related to sociodemographic characteristics were as follows: education, marital status, garbage disposal practices, age in years, place of temporary residence (in the plant area), and region of origin. Regarding risk behavior and health conditions, the following variables were considered: smoking status, history of hospitalization, and seeking health services in the last 12 months.

As some information in the database was not characterized as predictive variables, a brief description was made of the sanitary conditions, housing, and vaccine history of the sugarcane cutters enrolled in the study and who were seroreactive to hantavirus antibodies. In order to determine which of the 634 sugarcane cutters had been infected, their serum samples were screened using the ELISA IgG Andes immunoenzymatic assay, as previously described [16]. Samples that were reactive to a titration of 1:400 were subsequently tested at dilutions from 1:400 to 1:6400 to determine the maximum dilution with detectable anti-hantavirus IgG antibodies (endpoint). The prevalence of hantavirus infection was estimated, and the associations between the outcome variable (IgG anti-hantavirus reactivity) and the predictor variables were analyzed using a logistic regression model, which generated estimates of the prevalence odds ratios and respective 95.0% confidence intervals (95.0% CI). The variables (i) city of work (Americano do Brasil), (ii) history of vaccination for yellow fever, tetanus, and rubella (unknown), and (iii) region of birth (north; southeast) were excluded from the modeling due to low representativeness. Initially, a simple model was created for each variable using the chi-squared test (χ^2^), where *p*-values less than 0.05 were considered statistically significant (*p* < 0.05). Subsequently, multivariate logistic regression was used to generate the odds ratio estimates and respective 95.0% confidence intervals. Data were analyzed using IBM’s SPSS statistical software (version 20.0).

Graphs and tables were created using Microsoft Office Excel (2013), GraphPad Prism (version 5.0), and RStudio (version 1.2.5033). The figures presented in the results section were constructed using Microsoft PowerPoint (2013) and Photoshop (CS6 version 13). The Quantum GIS^®^ (QGIS version 3.32) program was used to generate distribution maps using the Datum WGS-84 geodetic system for all coordinates. The geographic locations of HPS cases were obtained based on the centroid of the notification municipality, due to the lack of information on the precise coordinates of the data taken from SINAN. The same procedure was performed for the cross-sectional seroepidemiological investigation data on hantavirus infection in manual sugarcane cutters in Goiás (see sections on methods and results). The geographical coordinates of the sugarcane plants were obtained from the lists of plants associated with SIFAEG (Union of the Ethanol Manufacturing Industry of the State of Goiás) and SIFAÇUCAR (Union of the Sugar Manufacturing Industry of the state of Goiás). The vectorized files were obtained from the Brazilian Institute of Geography and Statistics (IBGE), which uses the Datum WGS-84 in its cartographic bases. The raster layer on land use in the state of Goiás was obtained from MAPBIOMAS in the 2.0 collection [17].

### 2.3. Ethics Statement

This study was approved by the Research Ethics Committee of Hospital das Clínicas, Universidade Federal de Goiás (approval number 45109115.3.0000.5083), and the Fundação Oswaldo Cruz/Instituto Oswaldo Cruz Research Ethics Committee (approval number CAAE61629416.2.1001.5248).

## 3. Results

### 3.1. Epidemiological Situation of Hantavirus in Goiás According to Data Obtained from SINAN

In the state of Goiás, between 2007 and 2017, 95 cases were reported, with an average of 8 cases per year and annual numbers ranging from 3 to 14 cases. Regarding the municipalities of Goiás State, the largest number of cases was registered in Goiânia (*n* = 33, 34.4%), followed by Anápolis (*n* = 20, 20.8%) and Jataí (*n* = 11, 11.4%). In this case series, 67.5% were male, and the majority reported living in urban areas (75.0%), followed by rural (22.0%) and peri-urban (1.0%) areas; the remaining 2.0% were recorded as unknown. For the variable environment of infection, the home was mentioned in 26 of the HPS cases (27.0%), the workplace in 25 (26.0%), and leisure activities in 8 (8.3%). As for the age variable, in Goiás State, it is possible to observe a greater number of confirmed cases in the age group between 20 and 39 years old (*n* = 53, 55.2%), followed by 40 to 59 years old (*n* = 27, 28.1%).

Among the confirmed HPS cases in Goiás, 36.0% did not specify their education level, 24.0% studied until incomplete elementary school, and 22.5% had completed high school. Concerning skin color, the most affected in the state of Goiás were also individuals with brown skin (54.0%).

### 3.2. Epidemiological Characteristics of the Population of Sugarcane Cutters in Goiás

All of the sugarcane cutters were men (*n* = 634), with an average age of 35 years, and 71.0% reported being married (Table 2). Most of them had migrated from the Northeast region of the country. Almost half of the study participants lived in shared accommodation, and most individuals had access to a bathroom. As for water supply, half received piped water from a company, and most reported having garbage collection, while the others burned or buried their waste.

Regarding health-related behaviors, 80.3% of the workers had no history of smoking, 60.7% reported a previous hospitalization without details of the condition, and 59.8% had sought health services at some point in the last twelve months. Concerning vaccination history, the vast majority reported having received tetanus (83.1%), yellow fever (81.9%), hepatitis B (56.5%), and rubella (44.5%) vaccines.

Regarding housing, 43.0% of the total population lived in shared accommodation, the homes had up to 200 rooms, and the largest number of people in a home was 800. For water supply, 58.0% reported filtering, 1.0% boiled, 39.0% reported no treatment, and 2.0% were unable to inform. Among the seroreactive manual sugarcane cutters, it was also observed that all had a bathroom or toilet in their houses. Most of these seroreactive individuals lived in shared accommodation (57.0%). (Table 3).

### 3.3. Profile of Hantavirus-Seroreactive Individuals

Out of the 634 manual sugarcane cutters, 44 were found to have anti-hantavirus IgG antibodies in the 1:400 screening titrations, resulting in an overall exposure seroprevalence of 6.9%. Antibodies were also detected in nine samples at the 1:600 dilution and in three samples at the highest tested dilution (1:6.400).

Of the 44 seroreactive individuals, 48.0% worked in Serranópolis, 20.0% in Rubiataba, 18.0% in Anicuns, and 14.0% in Carmo do Rio Verde, all located in Goiás in the Midwest region, and 70.0% were married or in a stable relationship. Their ages ranged from 20 to 63 years, with an average of 35.5 years, and 57.0% had ≤5 years of education. Among them, 79.5% had garbage collection in their homes, while 20.5% reported burning their garbage. Of the total, 89.0% were originally from the Northeast region, and 11.0% were from the Midwest. The association of seropositivity (anti-hantavirus IgG) with most covariates was weak and not statistically significant, except for the city of work (marginally significant) and access to garbage collection (Table 4).

Although those born in the Northeast region showed a substantial association with seroreactivity (*p* = 0.13), and residents in the city of Serranópolis showed a higher prevalence than other cities (*p* = 0.05), these associations were not statistically significant.

Out of the 44 individuals, 16.0% had a smoking habit, and 59.0% reported having visited a health unit at some point in the last 12 months. Regarding the history of hospitalization, 68.0% confirmed having been hospitalized. Regarding vaccination history, 89.0% of the individuals had received the yellow fever vaccine, 54.5% had received the hepatitis B vaccine, 86.0% had received the tetanus vaccine, and 34.0% had received the rubella vaccine.

None of the health-related variables available showed any significant association with hantavirus seroreactivity (Table 5). Smokers was associated with lower seroprevalence than non-smokers (*p* = 0.51). It was observed that among those who had sought health services in the last 12 months (*p* = 0.92) and those who had been hospitalized previously (*p* = 0.30), the associations were not statistically significant.

### 3.4. Spatial Distribution of Hantavirus Cases in Goiás

According to data available on the SINAN website, the cases notified in Goiás from 2007 to 2017 were concentrated in Goiânia (41.0%), Anápolis (25.0%), Jataí (14.0%), Catalão (6.0%), Aparecida de Goiânia (5.0%), Cristalina (3.0%), Bonfinópolis (1.0%), Campo Alegre de Goiás (1.0%), Corumbá de Goiás (1.0%), Goianápolis (1.0%), Mineiros (1.0%), and Rio Verde (1.0%). The map in Figure 2 shows the locations of the cases registered by SINAN and the sugarcane plants where the cutters worked. It also indicates that Goiás State is mostly covered by forests and countryside formations, in addition to a mosaic of agriculture or pasture. Seropositivity to hantavirus antibodies appeared to follow the spatial distribution of agriculture and pasture. No reactivity was found among the analyzed samples in the municipality of Americano do Brasil.

## 4. Discussion

In recent decades, there has been an increasing investment in new alcohol-producing plants in several regions of Brazil, including the southwest region of the state of Goiás. This fact has resulted in the creation of new jobs for sugarcane cutters. However, the intensification of these activities, which require long working hours that often occur in unsanitary conditions, represents a health risk for these workers [18].

In this scenario, high unemployment rates in certain regions of Brazil compel young people to migrate in search of new job opportunities. This fact is likely related to the large number of Northeasterners among the sugarcane cutters included in the seroepidemiological survey of the present study. The Northeast region has been characterized by an intense migratory flow due to factors such as economic stagnation, social inequalities, and high levels of unemployment [19].

However, it is widely acknowledged that land-use changes and agricultural expansion—particularly sugarcane plantations—are manmade factors that can influence the spread and distribution of infectious diseases, particularly emerging zoonotic diseases such as HPS [20,21]. Muylaert and colleagues (2019) have shown that sugarcane plantations are among the most significant factors influencing the increased risk of HPS [22]. With the conversion of pre-existing pasture areas into sugarcane plantations, there is the possibility of an increase in the population of hantavirus rodent reservoirs. It is important to note that many of these rodents are generalist species that can adapt to different habitats, increasing the likelihood of encountering human populations, thus increasing exposure to the rodents’ excreta [23,24].

Analyzing the available data on the SINAN website reveals an increase in the number of HPS cases in Goiás State between 2007 and 2017. There was no significant seasonality in the distribution of deaths due to HPS in 2007–2015, except for a slight reduction in cases in February, November, and December, [25,26]. However, studies on the temporal trends of the disease have shown that the distribution of cases by region in Brazil reveals different patterns of seasonality [27,28]. Pinto and colleagues (2014) demonstrated a clear regional difference in the occurrence of the disease, with higher prevalence of HPS cases in the autumn–winter period in the Southeast and Midwest regions, and during spring in the states of the South region [27]. An important factor linked to seasonality is the higher reproduction rate of rodents during the hottest times of the year, generating large populations with an increased risk of spreading hantaviruses. The increase in rainfall is also associated with an increase in the supply of seeds for these animals, resulting in a population peak of hantavirus rodent reservoirs [29].

During the study period (2007–2017), the lethality rate in Goiás State was approximately 40.0%, which is similar to that seen in recent years. This rate is consistent with published Brazilian HPS lethality data, except for Santa Catarina, a highly endemic area where 29% lethality rates have been reported [30]. Also, HPS cases in Goiás State were mainly reported in men, consistent with the global literature, likely due to their occupational activity, which predominates within the agriculture sector [31,32]. The disease is commonly associated with rural environments [33], but it is important to note that individuals from urban areas were also affected, as we observed in this case series in Goiás State, where most of the cases reported living in urban areas. It is worth considering that despite living in urban areas, their residences may be located close to rural/wild environments or in places that were previously rural areas and have recently been occupied [34]. The probable places of infection most frequently reported were the home and work environments. In the working environment, infection is quite common due to activities such as cleaning warehouses and closed barns. In a cane field, rodents that feed on sugarcane may also be present and take advantage of food scraps that are often left on the ground by workers at mealtimes. Workers often get rest by lying down on the ground, and they can thus become contaminated when they come into contact with rodent feces and urine [35].

According to the IBGE microregion division of Goiás State, HPS cases are concentrated in Goiânia, followed by the southwestern municipalities of Goiás, namely, Jataí, and Anápolis. The presence of sugarcane plantations in Jataí reinforces the results obtained in the present study regarding the seroprevalence among populations of manual sugarcane cutters [36]. However, an important factor to consider is that there were no HPS cases reported by SINAN in other municipalities in Goiás State related to the investigated sugarcane cutters. Given the serological evidence of virus circulation in these municipalities with sugarcane plants, it is possible to question whether cases are being detected in these areas. Additionally, it is necessary to consider the possibility of misdiagnosis with other conditions—particularly dengue, which is more frequent and whose initial clinical manifestation can be very similar to the first stage of the disease [37,38]. Therefore, there is a need to implement epidemiological surveillance actions in these areas and provide a healthcare system that can serve vulnerable populations involved in growing and harvesting sugarcane, who are often exposed to environments shared with different reservoirs—both invertebrates and vertebrates—such as hantavirus-transmitting rodents.

Among the seroreactive individuals in this seroprevalence study, 59.0% had sought medical attention at a health unit in the last 12 months, and 68.0% reported hospitalization in this period. No statistical significance was observed regarding the number of times workers sought healthcare and the presence of hantavirus antibodies, or previous hospitalization history. However, it is important to note that hantavirus infection can be asymptomatic or oligosymptomatic, presenting as a flu-like condition without the need for hospitalization [39].

Although smoking has previously been significantly associated with the presence of hantavirus antibodies (adjusted OR 1.54; 95.0% CI 1.16–2.04), with significant dose–response relationships for the number of cigarettes smoked daily (OR 1.14; 95.0% CI 1.12–1.28) [40], this association was not observed in the present study. In some studies, smoking is considered to be a risk factor for HPS, suggesting that transmission occurs indoors and depends on the condition of the respiratory tract [41]. Bergstedt et al. [42] observed a significantly higher seroprevalence in smokers than in non-smokers (20.6% vs. 12.3%). However, the difference was not significant in the multiple regression analyses when adjusted for other variables.

Among the individuals who tested positive for hantavirus infection in the study, 57.0% lived in clusters. Despite the presence of running water and sewage treatment, the living arrangements and lack of hygiene in these homes can often lead to the accumulation of garbage, which attracts arthropods and vertebrates. In this scenario, it is possible to observe rodent infestations in homes where many people live, with rodents sharing space and food with humans [43].

Regarding the destination of garbage, the majority reported collection at home, which was the only statistically significant variable for hantavirus infection observed in the analyzed population. While not having garbage collection appeared to increase the risk of hantavirus infection, it is important to consider that information on the handling of garbage, the frequency of collection, the presence of rodents, the breeding of other animals for food, or the presence of small plantations in the surroundings of residences, among other factors, could not be evaluated. Given the high number of individuals in the same residence, for example, it is possible to question whether the accumulation of garbage due to delayed removal could attract rodents to the site, leading to an increased risk of contact with rodents and their excretions [44].

The overall seroprevalence of hantavirus infection in manual sugarcane cutters (6.9%) found in the present study was relatively high, in agreement with the SINAN data, which revealed a large number of cases among workers from rural areas, and with studies that highlight the importance of providing greater attention to this group of workers, who are often exposed to extreme conditions during their daily work [45,46]. It should be noted that other studies carried out in the state of Goiás, with about 400 participants, found lower seroprevalence rates of 2.6% in a rural population and 3.9% in the general population [47,48]. In a serological survey by Moreli et al. conducted in the municipality of Jataí [48], a seroprevalence of 3.9% (17/429) was identified, with higher prevalence among men (71.0%) and those living in rural areas (4.4%).

It is important to mention that the main limitation of this study was the use of secondary data on reported cases, which may have limited accuracy and lack variables of interest, such as occupation. The lack of comparison groups of other occupation or population categories also hindered the analysis of the additional infection risk posed by the work in sugarcane plantation and processing.

## 5. Conclusions

HPS cases in Goiás (2007–2017) were concentrated in young adult males, infected mainly at home and in urban areas, in contrast to the profile observed for other regions of Brazil, where rural environments are the main places of exposure to the virus. The highest number of cases occurred in the most populous cities of the state (Goiânia, Anápolis, and Jataí), and the number of confirmed cases per year fluctuated, in line with the fact that the disease occurs sporadically, influenced by the increase in the rodent population, which occurs in warmer times of the year.

Despite the absence of HPS cases reported in the municipalities where the sugarcane plants enrolled in the serosurveillance study (Serranópolis, Carmo do Rio Verde, Anicuns, Americano do Brasil, and Rubiataba) are located, we found the highest prevalence rates among young adults (62.5%), indicating exposure for those initiating their work life. Thus, the overall prevalence for hantavirus described here (6.9%—44/634) is within the range found for other rural populations from different regions of Brazil, consistent with the values found in other areas where the disease is endemic (0.0–21.2%). Nevertheless, not having garbage collection at home was significantly associated with the risk of exposure to hantavirus in our study, which indicates that these infections may not be related to sugarcane harvesting. Although, in our study, we were not able to identify where the exposure to hantavirus occurred, it is crucial to investigate possible cases in these cities with intense agricultural activities and a prominent sugar and alcohol industry. Therefore, prospective studies involving other vulnerable populations, such as sugarcane cutters in the Midwest region of Brazil, may increase our understanding of hantavirus infections and help reduce the invisibility of a highly lethal disease that currently has no specific treatment or vaccination.

## Figures and Tables

**Figure 1 viruses-15-02238-f001:**
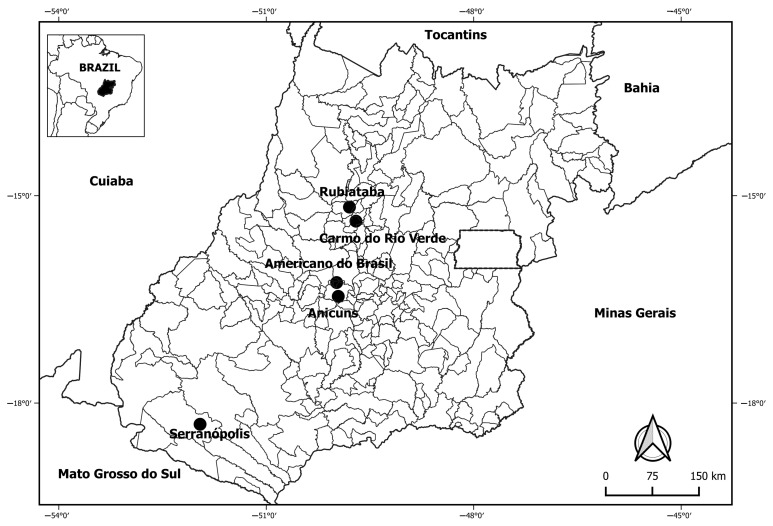
Locations of alcohol- and sugar-producing units where individuals were enrolled, according to municipalities of the state of Goiás, Brazil, 2016.

**Figure 2 viruses-15-02238-f002:**
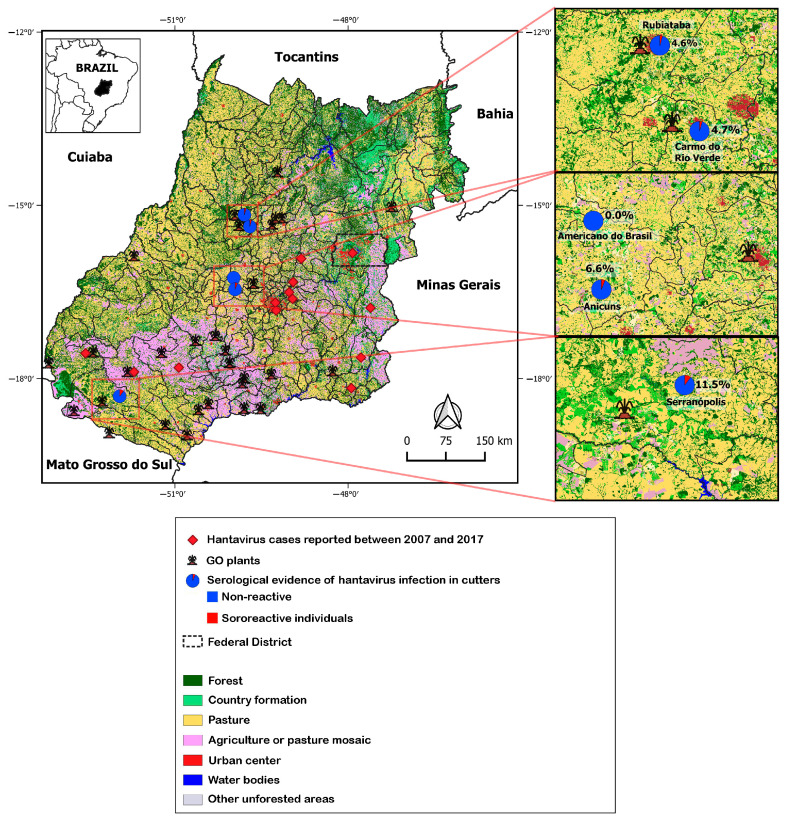
Spatial distribution of serological evidence of hantavirus infection in manual sugarcane cutters and hantavirus pulmonary syndrome cases in Goiás State (2007–2017), according to data from SINAN. GO: state of Goiás.

**Table 1 viruses-15-02238-t001:** Variables selected from data available on SINAN (Notifiable Diseases Information System) for descriptive analysis of the occurrence of HPS in the state of Goiás, Brazil (2007–2017).

Variable	Possible Answers
Environment of infection	Unspecified, home, work, leisure, and other
Confirmed cases per year	Number of cases per year (2007–2017)
Education level	Not specified, incomplete primary education, complete primary education, incomplete secondary education, complete secondary education, complete higher education, and incomplete higher education
Age group (years)	18–29, 30–49, 50–65
Notification municipality	Anápolis, Aparecida de Goiânia, Bonfinópolis, Campo Alegre de Goiás, Catalão, Corumbá de Goiás, Cristalina, Goianápolis, Goiânia, Jataí, Mineiros, and Rio Verde
Race	Unspecified, white, black, brown, indigenous, and yellow
Sex	Male or female
Residence area	Anápolis, Goiânia, surroundings of Brasília, other municipalities in the southwest of Goiás, Pires do Rio, Catalão, and Quirinópolis

**Table 2 viruses-15-02238-t002:** Sociodemographic characteristics of 634 manual sugarcane cutters from Goiás State, Brazil, 2016.

Sociodemographic Characteristics	*N* (Total)	%
Working city		
Americano do Brasil	7	1.1
Anicuns	122	19.2
Carmo do Rio Verde	128	20.1
Rubiataba	194	30.6
Serranópolis	183	28.8
Destination of the garbage *		
No collection	66	10.4
With collection	566	89.2
Marital status		
Married/stable union	452	71.3
Single/separated/widowed	182	28.7
Schooling (years)		
≤5	190	29.9
6–10	295	46.5
11–15	149	23.5
Age (years)		
18–29	189	29.8
30–49	396	62.5
50–65	49	7.7
Region of origin		
Midwest	129	20.3
Northeast	500	78.8
North	4	0.6
Southeast	1	0.1

* The assessment was only carried out on 632, due to a lack of information in the bank regarding the destination of the waste from 2 sugarcane cutters included in the study.

**Table 3 viruses-15-02238-t003:** Descriptive analysis of sanitary conditions, housing, and vaccination history of sugarcane cutters in Goiás State, Brazil, 2016.

Characteristics	Total Population (*n* = 634)	Hantavirus-Seroreactive (*n* = 44)
Bathroom/destination of human waste		
Bathroom	621 (98%)	44 (100.0%)
Septic tank	279 (44%)	22 (50.0%)
Rudimentary	184 (29%)	12 (27.0%)
Sewage	146 (23%)	8 (18.0%)
Straight into the river	6 (1.0%)	0 (0.0%)
Water supply		
Company plumbing	349 (55%)	16 (36.0%)
Piped from wells	108 (17%)	9 (20.5%)
Not plumbed	82 (13%)	10 (23.0%)
Piped from cistern	58 (9.1%)	5 (11.4%)
Pipeline from lakes	25 (3.9%)	3 (6.8%)
Shared housing	273 (43%)	25 (57.0%)
Number of people	1–800 people	1–430 people
Number of rooms	0–200 rooms	1–90 rooms
Water treatment		
Filters	368 (58%)	29 (66.0%)
No treatment	247 (39%)	12 (27.3%)
Boiling	6 (1.0%)	1 (2.3%)
Vaccination history		
Tetanus	527 (83.1%)	38 (86.0%)
Yellow fever	519 (81.9%)	39 (89.0%)
Hepatitis B	358 (56.5%)	24 (54.5%)
Rubella	282 (44.5%)	15 (34.0%)

**Table 4 viruses-15-02238-t004:** Associations between sociodemographic characteristics of manual sugarcane cutters and IgG reactivity to hantavirus in Goiás State, Brazil, 2016.

Variables	IgG-Reactive Samples (%)	IgG Non-Reactive Samples (%)	OR (95%CI)
Age			
<29	12 (6.3%)	177 (93.7%)	0.76 (0.23–2.48)
30–49	28 (7.1%)	368 (92.9%)	0.85 (0.29–2.55)
>50	4 (8.2%)	45 (91.8%)	1.0
School (years)			
<5	18 (9.5%)	172 (90.5%)	1.45 (0.65–3.25)
6–10	16 (5.4%)	279 (94.6%)	0.80 (0.35–1.80)
11–15	10 (6.7%)	139 (93.3%)	1.0
Marital status			
Married	31 (6.9%)	421 (93.1%)	0.95 (0.49–1.85)
Single	13 (7.1%)	169 (92.9%)	1.0
Region of birth			
Midwest *	5 (3.9%)	124 (96.1%)	1.0
Northeast	39 (7.8%)	461 (92.2%)	2.09 (0.81–5.43)
Work city #			
Anicuns	8 (6.6%)	114 (93.4%)	1.0
Carmo do Rio Verde	6 (4.7%)	122 (95.3%)	0.70 (0.236–2.08)
Rubiataba	9 (4.6%)	185 (95.4%)	0.64 (0.26–1.85)
Serranópolis	21 (11.5%)	162 (88.5%)	1.85 (0.79–4.31)
Garbage destination			
No collection	9 (13.6%)	57 (86.4%)	2.39 (1.10–5.23)
Collection	35 (6.2%)	531 (93.8%)	1.0

* The North and Southeast regions of birth were excluded from the analysis due to their low representativeness. # The municipality of Americano do Brasil (GO) was excluded from the analysis due to its low representativeness. OR: odds ratio, CI: confidence interval.

**Table 5 viruses-15-02238-t005:** Associations between health-related variables and reactivity to anti-hantavirus IgG among sugarcane cutters in Goiás State, Brazil, 2016.

Variables	Reactive Samples (%)	Non-Reactive Samples (%)	OR (CI 95.0%)
Smoking			
No	37 (7.3%)	472 (92.7%)	1.32 (0.57–3.04)
Yes	7 (5.6%)	118 (94.4%)	1.0
Sought health services in the last 12 months			
No	18 (7.1%)	237 (92.9%)	0.75 (1.51–2.76)
Yes	26 (6.9%)	353 (93.1%)	1.0
Previous hospitalization			
No	14 (5.6%)	235 (94.4%)	0.70 (0.36–1.35)
Yes	30 (7.8%)	355 (92.2%)	1.0

## Data Availability

The datasets supporting the conclusions of this article are included within the article and its tables.

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
