# Peer review of "Seroprevalence of Hantavirus among Manual Cane Cutters and Epidemiological Aspects of HPS in Central Brazil"

_viruses, 2023, doi:10.3390/v15112238_

Round 1

Reviewer 1 Report

The authors have determined the seroprevalence of hantavirus infections among sugarcane cutters, using the data and samples of a previous study. They also analysed the data of confirmed hantavirus pulmonary syndrome cases from the official surveillance database. The results add to the knowledge of the seroprevalence of hantavirus infections including risk factors. My comments are mainly related to the way of how the information is presented, as I got lost in the text several times.

Main comments

1.       Please consider rewriting the abstract. It was very confusing for me to read that the aim was to investigate the seroprevalence among sugar cane cutters, but in the methods that a retrospective study of all confirmed cases was done, with in the results the results of the seroprevalence study. After reading the rest of the manuscript, I could understand what the authors meant in the Abstract. However, an abstract should be understandable independent of the article.

2.       Please consider rearrange some sentences in the introduction. By moving lines 48-54 (‘Transmission…nose [8,9]’) to line 45 (after ‘…rodents,’) a more natural flow can be created.

3.       What is meant by ‘The analysis took into account the survey’ (2.1, line 78)?

4.       Lines 84-86. The deaths are extracted from another database? If yes, could these data be linked to ascertain the status of the individual cases?

5.       Please consider moving the lines 87-102 (‘Graphs…collection [16].’) to an own paragraph before 2.3 Ethical Statement. Furthermore, please specify which municipality (notification or residence) is meant with ‘centroid of the municipality’ (line 92).

6.       In line 145, the authors state that there were 95 cases (2007-2017), whereas in line 164 this is 76 cases (2007-2017)?

7.       Lines 222-224 and Table 5. I think that ‘A higher prevalence’ is not accurate here. First, 6.9% is lower than 7.1% and second, p is 0.92 and thus (very) far from statistically significant (besides the already very small difference of 0.2%).

8.       Overall, the discussion was difficult to follow, as different topics are discussed without a clear flow. Try to connect the topics more, maybe rearranging some parts, and thus creating a more coherent discussion. Some specific points:

a.       Lines 254-261 (‘However…excreta [23,24].’) and lines 310-317 (‘Muylaert….disease [43,44].’) have a lot in common, partly even repetitive. Similar accounts for lines 285-294 (‘The probable…urine[38].’) and lines 333-338 (‘Among…humans [48].’)

b.       Lines 262-273: in Goias no seasonality is seen, but in Brazil is including factors linked to it? What is the message you want the reader to remember on this topic?

9.       Conclusion. In the conclusion, no answer is given on the aim of the study. What were the main results, what could we learn/should we have learned from this study, what knowledge did it add?

Minor comments

10.   Throughout the text both ‘sugar cane’ and ‘sugarcane’ is used. Please choose one of them and use it consequently.

11.   Please be consistent with the number of decimals given when presenting percentages. A guideline could be no decimals when denominator is 100 or less and one decimal when the denominator is larger.

12.   Line 51, ‘eliminated’. I think ‘excreted’ is meant?

13.   Table 1, variable ‘confirmed cases per year’. Should that not be ‘year of confirmation’, as the possible answers are ‘2007-2017’?

14.   Line 146: the number of cases increased in 2009, 2013 and 2015. However, if understand correctly there were 33 cases in total in 10 years time, which would be an average of 3 per year. Thus, please, specify the increase.

15.   Lines 173-175. I assume that previous hospitalization is without a defined time period as the percentage is higher than that of seeking health services in the last twelve months?

16.   Table 4. The row of ‘<5’ has been shifted. It is now below age, but belongs to the variabel school. Furthermore, in the last column, some ‘,’ for decimals have to be replaced by ‘.’

17.   Line 358, 70%. This percentage seems out of order, as all other seroprevalence percentages given in the surrounding sentences are below 5%. Please check.

18.   References. Several references are incomplete (no journal issue, page number, etc, for example nr. 12 and 21) or parts are in Portugese (for example nr. 1 and 14). And one article is referenced twice (50 and 53).

An English language check would improve the readability of the manuscript as I think there are several minor grammatical errors in the text.

Reviewer 2 Report

The major problem with this serological study is that the presence of antibodies does not indicate when or where the positive individuals were infected. Apparently, most of the 44 positive individuals were from the northeast, outside of Goías. The authors appear to presume that these seropositive individuals were infected in the areas where they were sampled in Goias. If the authors have evidence of this they need to state that. If no evidence is available, the relationship of infection to environmental and occupational risk should be reconsidered and the discussion and conclusions of the study need to be reorganized.

Line 23. ...prevalence of hantavirus antibodies.

Line 28. ...tested for hantavirus antibodies... not the virus itself.

Line 66, 70. ... cases of hantavirus infections.., Were these all HCPS cases or were some just febrile or even asymptomatic? If so, how many of each?

Line 68. The case definition used in this study needs to be clearly defined. ... we assessed the epidemiological profile of reported previously infected  hantavirus cases in Goiás...Since antibody just indicates past infection, it says nothing about the severity of the infection other than it was not fatal nor where the infection was acquired. Line 168 indicates that most cane cutters were from the northeast. Line 152 and 156 imply that all were HPS cases.  Is this correct?

Line 91 HPS cases

Line 161. "the hantavirus" implies that only one hantavirus was involved. That is likely not the case in all of Brazil. It would be safer to state" hantavirus infections."

Line 122 In order to determine which of the 634 sugar cane cutters had been infected, their serum samples were screened...

Line 145. HCPS cases? Cases of past hantavirus infection? Also, use either HPS or HPCS throughout.

Lines 339-348. This paragraph presumes that the hantavirus antibodies detected were the result of infection in the area where the 44 positive individuals were sampled. How do the authors know that? They state that most of the cane cutters were from the northeast. How do they know that their infections were or were not acquired there?

The English is quite good and just a few changes by the copy editor are needed. 

Round 2

Reviewer 2 Report

The earlier criticisms have been adequately addressed.

There are very few errors. The writing is clear.